# Vegetated Roofs as a Means of Sustainable Urban Development: A Scoping Review

**Mohammad A. Rahman, Mohammad A. Alim** **, Sayka Jahan and Ataur Rahman ***

School of Engineering, Design and Built Environment, Western Sydney University, Office: XB.3.43, Kingswood (Penrith Campus), Locked Bag 1797, Penrith South DC, NSW 2751, Australia
* Correspondence: a.rahman@westernsydney.edu.au

**Abstract:** Urbanisation affects the water cycle and heat balance in a negative way. Vegetated roofs have the potential to minimise the effects of urbanisation. A scoping review is presented here to examine how vegetated roofs are being evolved as an effective tool of sustainable urban stormwater management and overall urban development. It has been found that research on vegetated roofs has been increasing significantly and it can contribute towards achieving multiple sustainable development goals (SDGs). It has also been found that the uptake of vegetated roofs has been slow. A lack of regulatory acceptance caused by an absence of experimental data and a subsequent knowledge gap establishing the effectiveness of vegetated roofs are major reasons behind this slow uptake. Future research on vegetated roofs and their subsequent evolutions should put a focus on gathering experimental data towards establishing a performance benchmark for detention, retention and water quality in urban settings. Such data can be utilised towards developing a stand-alone guideline and software for green roof design.

**Keywords:** urbanisation; SDG; green roof; retention; detention; purple roof

## 1. Introduction

Currently four billion people live in urban areas in comparison to two billion in 1985 [1]. Such numbers are forecasted to rise to 6.68 billion by 2050, which is 68% of the world's population. This rapid urbanization coupled with climate change effects has led to several primary adverse effects such as waterlogging and flooding, Urban Heat Island (UHI) effects, loss of biodiversity and air pollution. These primary effects result in negative impacts on the physical and mental health of urban residents [2].

The land required for urban development is obtained by destroying forest, agricultural and farming lands. This process adversely affects the catchment hydrology by converting a permeable surface to an impermeable one and reducing evapotranspiration. Razzagh-manesh et al. [3] reported that between 62% and 90% of rainfall becomes runoff from conventional rooftops, with the runoff percentage increasing with tiled and higher degrees of roof slopes. These urbanisation-induced changes in land use/land cover increase the potential for pluvial flash flooding [4]. Such floods have resulted in significant economic losses and casualties worldwide over the past two decades [5]. In China, urban pluvial flash floods have been reported to cause a significant loss of lives and damages to property in megacities such as Shanghai, Wuhan, Shenzhen, Tianjin and Beijing [6].

In terms of water quality, approximately 30 to 50% of the world's surface water has been affected by urban stormwater or other types of pollution since the last century [7]. Non-point source pollution in conjunction with pluvial flooding and combined sewer overflows are reported to be a major threat to human life and property [8]. The heavy metal and polycyclic aromatic hydrocarbons (PAHs) associated with stormwater can cause acute toxicity and may be carcinogenic to humans—salt components such as bromide and chloride can disrupt the nervous system and nitrogen and phosphorus can cause the eutrophication of water bodies [9].

Green infrastructure not only provides stormwater management and air quality improvement, but also provides social benefits which are not easily quantified, such as community cohesion, stress and anxiety reduction and educational benefits [10–13]. Green infrastructure is also demonstrated as a key to improved public health, as evidenced by recent medical research which suggests that green infrastructure contributes positively to a stronger immune system [14]. A schematic of available green infrastructure types and their effectiveness are shown in Table 1.

**Table 1.** Green infrastructure practices and benefits.

| Practice | Stormwater Management | | Community Benefits | | | | | | | Footprint |
|---|---|---|---|---|---|---|---|---|---|---|
| | Improves Quality | Reduces Quantity | Improves Air Quality | Reduces Atmospheric $CO_2$ | Reduce Urban Heat Island Effect | Increase Recreational Opportunity | Improves Community Cohesion | Urban Agriculture | Improves Habitat | Minimal/ Retrofit |
| Green Roof | ✓ | ✓ | ✓ | ✓ | ✓ | ✓ | ✓ | ✓ | ✓ | ✓ |
| Tree Planting | | ✓ | ✓ | ✓ | ✓ | ✓ | ✓ | ✓ | ✓ | ✗ |
| Bioretention | ✓ | ✓ | ✓ | ✓ | ✗ | ✗ | ✗ | ✗ | ✗ | ✗ |
| Permeable Pavement | ✓ | ✓ | ✗ | ✗ | ✗ | ✗ | ✗ | ✗ | ✗ | ✓ |
| Water harvesting | ✗ | ✗ | ✗ | ✗ | ✗ | ✗ | ✗ | ✓ | ✗ | ✗ |
| Stormwater basins | ✓ | ✓ | ✗ | ✗ | ✗ | ✗ | ✗ | ✗ | ✗ | ✗ |

Increased urban areas are resulting in heat island effects across many cities [15]. For example, AlDousari et al. [16] examined the impacts of changes in land use/land cover on land-surface temperature and urban heat island effects in Kuwait using artificial neural network and support vector machine models; they showed that increased urbanisation would intensify heat island effects. Rahaman et al. [17] adopted a support vector machine and cellular automata algorithms to examine the impacts of land-use change on urban heat island effects in Penang, Malaysia. Kafy et al. [18] noted that in Dhaka city of Bangladesh, the increased urbanisation from 2020 to 2030 will result in higher summer and winter temperatures by 13% and 20%, respectively. Dey et al. [19] noted that a significant increase in urban density and decrease in green cover and water bodies within Rajshahi city in Bangladesh will notably increase the temperature and heatwaves by 2040. Kafy et al. [20] noted that an increase in urban areas is causing reduced vegetation, which is resulting in a higher land-surface temperature.

Nearly 50% of the total impervious area in highly urbanised localities is comprised of building roofs [21]. While the increased trend of urbanisation and subsequent addition to the impervious surface by building roofs are inevitable, there has been a pursuit to explore ways to mitigate their adverse effects. Converting concrete and tiles roofs into green roofs can increase green urban areas [15] and has the potential to reverse such effects. The unique beauty of green roofs is that it can efficiently use the existing building footprint to improve the urban green area [22]. Green infrastructure has also been reported to have significant positive impacts on the reduction in urban heat island effects [23].

The second largest problem from the concrete jungle we have created is the urban heat island effect. There is a desperate search of strategies that can mitigate urban heat island effects. Barnhart et al. [24] reported green roofs to be the most practical and popular type of

green infrastructure that can be implemented in highly urbanised watersheds due to their low cost and efficient utilisation of unused or under-used space.

Irrespective of these widespread benefits and global adoption of green roofs, Australia has been slow to adopt this technology [25]. Many of the environmental and economic benefits of green roofs are location-specific. Australia is a continent and there are significant spatial differences in rainfall, temperature, available substrates and suitable vegetation across Australia that can be used to build green roofs [26].

Williams et al. [26] listed several benefits of green roofs in an Australian context—namely, stormwater retention, building cooling benefits, building energy savings, potential to mitigate urban heat island, assist in wellbeing, increased biodiversity and improved air quality. They have also created a roadmap for wide green roof implementation in Australia, which comprises of gathering and sharing knowledge, collaborating and advocating, government coordination and national leadership, developing and implementing policy mechanisms, building skills and expertise, and finally, designing for success—for example, maintenance, integrating functions and farming. In addition to this, they have identified research gaps in green roof maintenance and management, the value of green roofs for local biodiversity, the potential to mitigate urban heat island, design and experience in the industry and innovative designs that integrate and optimise different green roof functions.

There is a lack of research and design guidelines on green infrastructure in Australia and in many other countries. To fill this knowledge gap, this scoping review has been undertaken with an objective to benchmark global green infrastructure-based research and to identify future challenges and opportunities to enhance the application of green infrastructure in urban areas. The innovation in this study is to identify how the green infrastructure can contribute towards more sustainable urban stormwater management, urban development and how it can fulfil the sustainable development goals.

## 2. Material and Methods

To undertake this review, we have followed a framework of a scoping review recommended by several researchers [27–29]. We have chosen one of the models where investigators map the research in a specific field and identify research gaps.

The first step is to formulate research questions, followed by the identification of keywords. The thought process that proceeded in this literature review included: "Can green roof system and its subsequent evolution sustainably meet the stormwater management challenge in urban development in Australia?" If the answer to the primary question was yes, then the secondary question was "Can green roof assist in meeting multiple sustainable development goals?" A range of additional questions were asked to fully understand several aspects of green roof systems such as:

1. What are the key challenges for stormwater management in Australia?
2. How has stormwater management evolved in Australia over time?
3. Can green roofs assist in meeting the challenges in urban development?
4. Where are the current knowledge gaps?
5. What are the future challenges and research opportunities?

After defining the research questions, we identified relevant keywords for searching and selecting the studies related to green roof systems. The following keywords were used to locate articles from the scientific databases: Water Sensitive Urban Design (WSUD), Integrated Water Cycle Management (IWCM), urbanisation, Sustainable Development Goals (SDGs), green roof, purple roof, standards, stormwater management, adverse effects of stormwater, retrofitting WSUD, sustainable development goals and green roof and knowledge gaps in green roof.

The scientific databases such as Scopus, Web of Science, Science Direct and Google Scholar were used to gather relevant publications. The next steps involved the selection of articles and the formation of the data bank with the selected articles. We focused on WSUD, IWCM, green roofs and their evolutions, financial feasibility and the knowledge gap in the green roof industry. More than 500 articles popped up in our literature search; however, we

selected only 71 articles based on our above-defined criteria. The final step was the analysis and comparison of accumulated findings and observations and the dissemination of the results in the form of this journal article.

## 3. Green Roofs as Means of WSUD

With the existence of rapid urbanisation, permeable surfaces are getting lost by industrially transforming permeable surfaces (i.e., grassland, forests and farms) into impermeable ones (i.e., residential, commercial and industrial spaces) [30]. In this process, native vegetation is being reduced or eliminated, the shallow depression of the natural soil and native drainage patterns that allow stormwater infiltration are being limited [31] and surface water intake for evapotranspiration by plants are being reduced. Consequently, a considerably increased amount of rainfall is being converted into urban runoff, leading to a higher flood risk [32].

Almost 50% of the world's population live in cities, which is expected to rise to 68% by 2050 [33]. Sustainable Urban Drainage Systems (SUDS) are considered essential to minimise the urbanisation impacts on hydrology and increase the resilience of urban centres to extreme rainfall events [34], as these facilities have the ability to attenuate rainfall [35].

WSUD, SUDS and Low Impact Designs (LIDs) (these are essentially similar) are known to provide multiple environmental benefits [29], such as:

1.  The capability of reducing the effects of extreme rainfall events mentioned above [36];
2.  Retaining particulate stormwater contaminants, such as TSS and particulate metals [37];
3.  Reducing the significant concentration of pollutants [38] and minimising the impacts of polluted water discharge to waterbodies [39];
4.  Removing microplastics [40];
5.  Providing thermal insulation of buildings;
6.  Improving microclimates and aesthetics, and hence, improving human wellbeing [41];
7.  Reducing the risk of flooding, waterlogging and maintaining the baseflows of receiving rivers in cities [41];
8.  The capability to act as a source of water [42]; and
9.  Reducing the combined sewer overflow volume and frequencies [43].

It seems crucial that a larger runoff reduction from disconnection strategies is achieved [44]. In adopting source control, well-established WSUD methods include permeable pavements, swales, bioretention trenches, infiltration trenches, tree pits, detention basins and wetlands. Joshi et al. [45] advised that whereas a lack of available land and constraints by existing infrastructures such as buildings and roads in highly developed areas make the adoption of WSUD incompatible, green roofs may overcome this constraint since they can be retrofitted within the existing building footprint by utilising hard to access and unutilised spaces, such as urban roofs. While doing so, they can provide numerous environmental, economic and social benefits [41] as they retain, detain and slowly release rainwater without the requirement of using new space [46].

Green roofs have also been reported to reduce greenhouse gases [47]. Plants reduce the surrounding temperature and increase the air humidity by transpiration [48]. This is recognised by several novel WSUD measures, such as sponge city in China, which acknowledges green roofs as a key component to sustainable stormwater management. Green roofs are also found to reduce combined sewer overflow frequency by 62% and volume reduction by approximately 20% [49]. Green roofs improve air quality [50] and may also help to mitigate climate change via carbon sequestration [51]. Green roofs' suitability as a WSUD component is presented in Figure 1.

| WSUD | Reduce Flooding | Attenuate Ruoff | Retention | Improve Quality | Reduce load to waterways | Thermal Insulation | Reduce CSO | Maintain baseflow of waterways |
|---|---|---|---|---|---|---|---|---|
| Green Roof | ✓ | ✓ | ✓ | Refer Note | ✓ | ✓ | ✓ | ✓ |
| **Note:**<br>Green Roof was found to remove microplastics, heavy metals and PAHs, but increased phosphorus and nitrogen concentration has been observed. | | | | | | | | |

**Figure 1.** Green roof's suitability for WSUD.

## 4. Evolution of Vegetated Roofs

A bibliometric analysis using VoSViewer and Citescape software for green roof research has been performed. The geographic distribution of this research is presented in Figure 2. Here, a bigger circle indicates a higher research concentration and connecting lines indicate research collaborations. It can be seen that European countries have dominated research on vegetated roofs, where Spain, Italy, Germany, the Netherlands, England, France, Turkey, Belgium and Denmark play a larger role. Beyond Europe, China, the USA, Canada, Japan, Australia, India and South Korea are prominent in this research area. In terms of networking, the USA, China, England, Italy and the Netherlands are at the top.

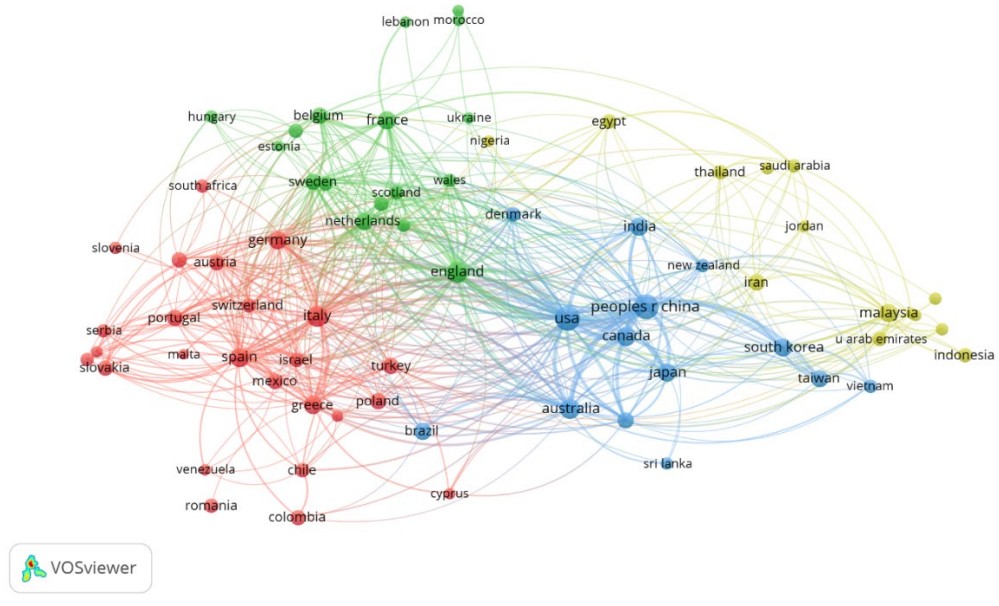

**Figure 2.** Geographical Distribution Network for Research on Green Roof Analysis.

Keyword analysis is presented in Figure 3. It can be seen that green roof/green roofs are the most frequently used keywords (represented by the biggest circle size). The next four most frequently used keywords are urban heat island effect, sustainability, green infrastructure and stormwater management. There are some more keywords which are closely connected with these groups, such as runoff, stormwater, water retention and water quality, of which are connected with stormwater management.

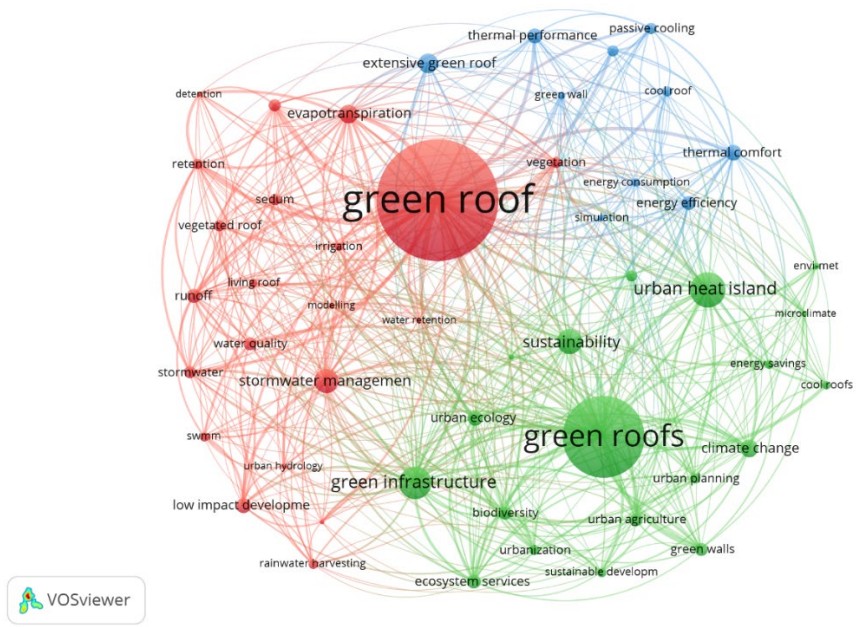

**Figure 3.** Keywords Review in Green Roof Research.

A green roof is generally comprised of a plant growth layer, followed by an imperme-
able layer. Although this system provides some benefits, the results are highly variable, as
observed by a significantly varying peak runoff delay. Joshi et al. [45] reported peak runoff
delay ranging from 0 to 30 min. Moran et al. [52] showed a peak runoff delay of approxi-
mately 4 h. In terms of peak flow reductions, Getter et al. [53] showed a considerable peak
reduction for low intensity rain. Stovin et al. [47] observed a mean peak flow reduction of
60% for rainfall events of more than 22 mm of accumulated rain, whereas Moran et al. [52]
reported a 90% reduction in peak flow.

## 5. Design, Construction and Maintenance of Green Roofs

Simmons et al. [54] suggested that "vegetated roofs are increasingly being incorporated
as a sustainable practice in building design, often without specific attention to designing
the roof to achieve specific functions, or to the conditions of a specific climate".

According to the Technical Guidelines by Inner West Council (2022) of New South
Wales, Australia [55], there are four key steps towards green roof construction: complete site
analysis, identification of opportunities and constraints, plan and design, and construction
and maintenance. These topic areas along with other key design and maintenance issues
are discussed below.

i.    Complete Site Analysis and Review Site Analysis to Identify Opportunities and
      Constraints

Site analysis is the most critical factor in designing a green roof. The regional, land-
scape and site scale need to be considered for the selection of the right type of green
roof [56], followed by a thorough review considering plant types based on sunlight and
water requirements and cold tolerance [57].

Secondly, properties of new or existing structures on which the green roof is to be
built need to be investigated. This should consider the size and slope of green roofs and
whether there are any existing plants or equipment, which then leads onto the structural
loading. Depending on the budget and loading, there may be an opportunity of integrating
solar panels with green roofs. Thirdly, access and constraints for both during construction
and long-term maintenance need to be considered. Maintenance is required for plants,
irrigation systems, structures and drainage systems. Fourthly, reviewing council planning
maps is necessary to identify any heritage requirements of the site. Fifth, water aspects
including drainage and irrigation requirements need to be looked at, and lastly, biodiversity,

flora and fauna need to be carefully considered. Recently, a law was passed in France mandating new rooftops in commercial buildings to be covered either in greens or solar panels; in 2014, the city of Sydney adopted the green roof and walls policy [58].

ii.　　Plan and Design

Specialist consultant advice should be sought for the design and documentation of green roofs. Depending on the size of the project, the engagement of a project manager, architect, landscape architect and a civil and structural engineer, and specialist consultants may be required. The benefits of using suitably experienced consultants on green roofs will generally result in a smoother, straight forward design, approval and construction process.

Secondly, local planning requirements and building standards need to be looked at and design objectives including water retention, detention, quality and thermal efficiency need to be considered; in addition, building rating systems need to be considered, irrigation and drainage systems should be thoroughly investigated and maintenance and access need to be planned. Finally, options for co-locating sustainable energy systems need to be considered. Additionally, careful consideration is required for the cost and use of recycled materials.

Design of a green roof system can largely be subdivided into the following categories:

iii.　　Plant Selection

Plant selection and substrate types and depths should be carried out from locally available guidelines. Plant selection depends on several factors including site conditions and design objectives. Plants adopted on extensive green roofs are typically shallow and fibrous-rooted, and are types of succulents and grasses. Extensive knowledge on plant selection is critical to the design and survival of successful green roofs [59]. Across North America, succulents were found to be the most successful plant species to be able to survive on the harsh roof-top environment.

Sedum showed very successful growth in most states of the US, whereas mixed results were observed in a few areas. Where Sedum roots suffered from low temperature and freezing at some locations, they performed poorly in hot and humid regions [60].

iv.　　Green Roof Design Elements and Components

The key component—apart from the substrate depth—is the growth media characteristics: materials, distribution of granular particle size, organic and nutrient contents, pH and permeability rates. Further research is required on the type of fabrics.

v.　　Growth Media Type and Thickness

Succulents were found best on 7 to 10 cm of media but were found to tolerate 5 cm depth as well. In another eco-region, plants sustaining on 2.5 to 5 cm shallow substrates were also reported [61]. Geosynthetics were found to be as critically important for managing moisture [62]. Grass and herbaceous green roof covers generally needed a substrate depth of 15 to 20 cm.

vi.　　Construction

A safety in design exercise must be completed considering the following Australian Acts:

- Work, Health and Safety Act 2011; and
- Work, Health and Safety Regulation 2017.

Risks and hazards during and after construction must be identified and designed out as far as possible during the concept and detailed design phases. During construction, waterproofing is one of the most important aspects. The growing medium should possess the following characteristics: efficient moisture retention, well aerated and drained, ability to absorb and supply nutrients, provide anchorage for plants, be lightweight and fire-resistant. The construction should be carried out by suitable professionals experienced in constructing green roofs.

vii.　　Use and Maintenance

As roof tops are not often visited by humans, green roofs are found to be a good habitat for birdlife; in addition, many species of spiders and butterflies are often observed. Two branches of knowledge are extremely critical for the selection of the right plant type: horticulture and ecology. Horticulture could be useful for selecting the right plant type, whereas ecology could be useful for setting up the right symbiotic relationship.

A suitable maintenance plan for the green roof needs to be designed and adhered to for the successful implementation of the green roof. The maintenance regime can be broken down into four phases: establishment maintenance, routine maintenance, cyclic maintenance and reactive/preventative maintenance. To ensure success of the green roof, plants need to receive proper levels of fertiliser depending on the requirements of each species. For larger projects, warranty and guarantee from the manufacturers, suppliers and installers and a suitable defect liability period need to be selected.

viii.   End of Life Scenario

Besides the critical factors for establishing a successful green roof, it is also equally important to consider the end-of-life scenario [63]. The green roof soil layers may be reused for agriculture. As per Butler et al. [64], incineration is recommended to be excluded due to the presence of a large amount of inert material. The potential presence of peat in green roof materials makes landfill a suitable disposal location of green roof materials. The end-of-life cost and management involve three steps: identification of products involved, identification of potential waste scenario and determination of cost hypotheses for each case study [65].

**6. Innovation**

The effectiveness of the green roof depends on the composition and properties of layers and their interactions [66]. To achieve the full benefit of the green roof and quantity of the retention and detention, the recent development enables an effective stormwater management by combining detention and retention in rooftop stormwater by adding two additional layers of mineral wool and honeycomb detention layer. The honeycomb detention layer has been varied by a buffer layer and such roofing mechanisms are called blue-green layers [67,68]. Figure 4 shows the key differences between a traditional green roof and purple roof.

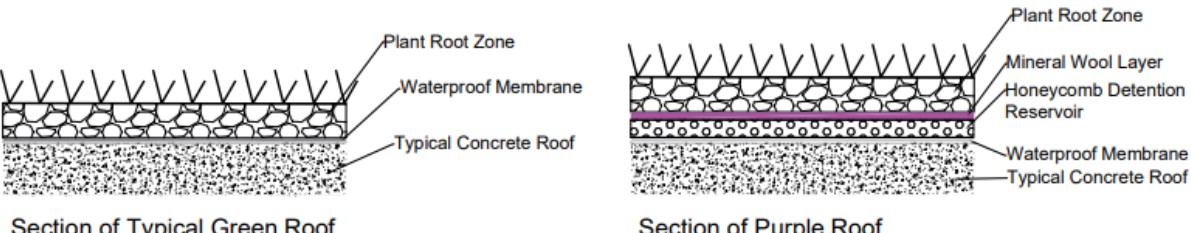

**Figure 4.** Typical Sections of a Green Roof and Purple Roof.

As observed, the first key difference between the traditional green roof and purple roof is the use of mineral wool as a detention and drainage layer. It is a flexible layer comprised of vertically oriented nylon threads between two knit layers of enclosing nylon fabric; this allows the drainage of the green roof assembly at a pre-determined flow rate, which creates stormwater detention and reduces stormwater peak flow and/or delay stormwater peak flow above a given flow rate, in accordance with the project stormwater calculation [67].

The second key difference for the purple roof is the use of the honeycomb detention reservoir. These are rigid sheets formed of fused polypropylene tubes oriented vertically when the sheet is laid horizontally. The tube diameters are generally 10 mm, with a thickness of 25 mm and provides a void space of >93% [63]. From a modelling exercise by Busker et al. [64], it was reported that without the blue layer, the green layer would store approximately 30% of the total precipitation with the balance overflowing, whereas with

the addition of the blue layer without any controlled drainage, the capture ratio increases to 50%, and with the addition of smart control, the capture ratio increases to greater than 90%.

## 7. Sustainable Development Goals vs. Green Roofs

At the Earth Summit in Rio de Janeiro in 1992, the United Nation's Sustainable Development Goals (SDG)s were first proposed. In 2015, with more than 12+ years' work by several countries, all the UN member states adopted "The 2030 Agenda for Sustainable Development". At the core of it, there are 17 SDGs that recognise the fact that ending poverty and other deprivations must go hand in hand with strategies that improve health and education, reduce inequality, and spur economic growth while tackling climate change and working to preserve the ocean and forests [69]. City of Sydney [58] have analysed the fit of green roofs with SDG and have concluded that green roofs satisfy 11 of these SDGs. We have reviewed this and concur with such findings with one additional SDG satisfaction—SDG 14: Life Below Water. A brief presentation of the green roof fitting in with the SDGs is given in Figure 5.

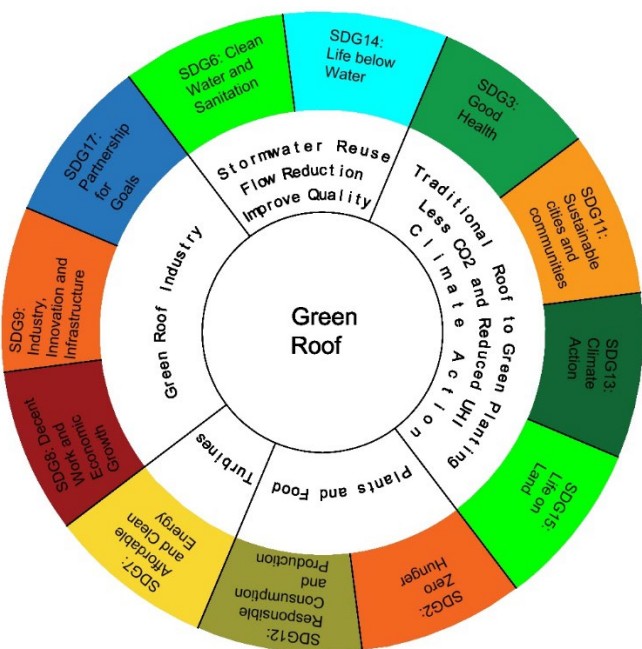

**Figure 5.** Sustainable Development Goals and Green Roof.

In recent studies, water retention was evident in the blue water storage layer on green roofs [68]. With the reduced volume, even at the same concentration, there was less pollutant load to the receiving water bodies [23]—this reduces the risk of eutrophication and assists in SDG 14: Life below water [69].

Harvested rainwater can be reused for non-potable domestic uses, such as toilet flushing and washing. This, in turn, reduces the pressure on both clean water supply demand and produces less loads on the sanitary system (SDG 6: Clean water and sanitation). In addition to this, such water quantity and quality benefits limit eutrophication, assist in maintaining the quality of receiving water bodies and support life such as phytoplankton and fish, hence, green roofs support SDG14.

The plants provide mental health benefits (SDG3: Good health and wellbeing). While doing this, green roofs provide a shelter for insects, bees and pollinators (SDG15: Life on land), and plants provide oxygen and remove carbon-di-oxide (SDG 13—Climate Action). Large scale implementation of green roofs could add significant green looks to the urban areas and create sustainable and resilient urban areas (SDG 11: Sustainable cities and communities). Multi-layered green roofs also have the potential to contribute to the source of food by implementing agriculture on green roofs (SDG 2: Zero hunger). This also

corresponds to SDG 12: Responsible consumption and production. Green roofs offer the potential to locally grow certain fruits and vegetables (SDG2: Zero hunger, SDG12: Responsible consumption and production). Water stored in large green roofs and its release has the potential to generate power through the use of turbines. Additionally, green roofs reduce the urban heat island effect and reduce the energy requirement for cooling (SDG 7: Clean energy).

Local food production reduces the need for lengthy transports and limits $CO_2$ emissions. This also corresponds to the reduction of energy consumption in cooling and heating and contributes to SDG 7: Affordable clean energy. Developments in the green roof industry are well aligned with the UN's Global Green New Deal Policy (GGND), which encourages the sustainable development of the economy to create new job opportunities and facilitate economic growth (SDG 8: Decent work and economic goals). While doing so, partnership and collaboration at both national and international levels become crucial for sharing knowledge and improving green roof technology. Such collaborations can happen on several areas, such as exploring new materials and techniques, layering, hybrids such as combination of solar panels [70] and, hence, can contribute to SDG 17 (SDG 17: Partnership to achieve goal). Continuous innovation and finding new materials and techniques promote SDG 9: Industry, innovation, and techniques.

## 8. Identification of Knowledge Gaps and Future Research Needs

The green roof is a relatively new technology. The first published article on green roofs was in 1960, then it began gaining momentum from 2009 (i.e., more than 100 published papers a year) and has gained widespread attention since 2017 (more than 400 published papers a year), as evidenced from the chronological arrangement of papers shown in Figure 6. Although green roofs have been widely used in Germany, and then the US, there is significant knowledge gaps in certain areas. To identify the knowledge gaps, a subjective analysis was performed on global knowledge and the US, and then in Australia, as presented in Figure 7.

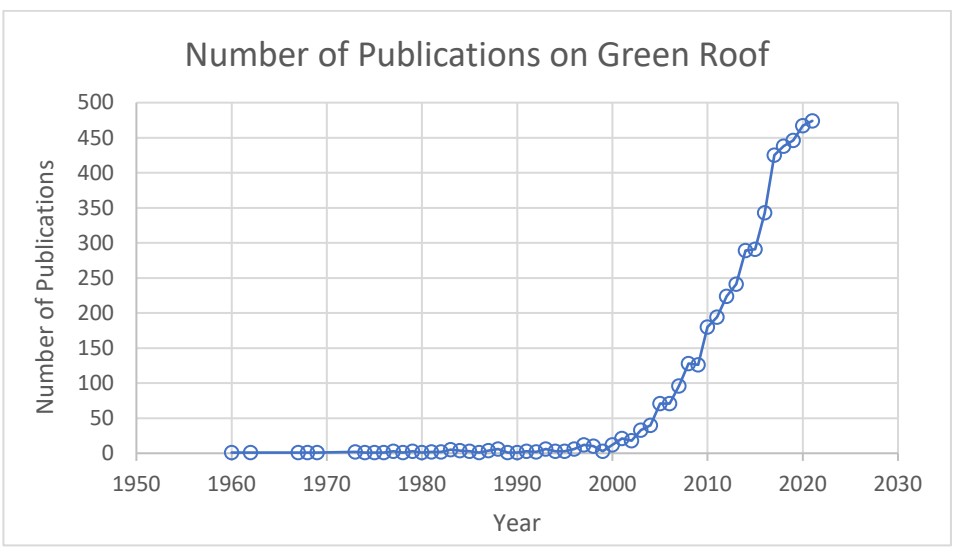

**Figure 6.** Chronological Assembly of Published Articles on Green Roof.

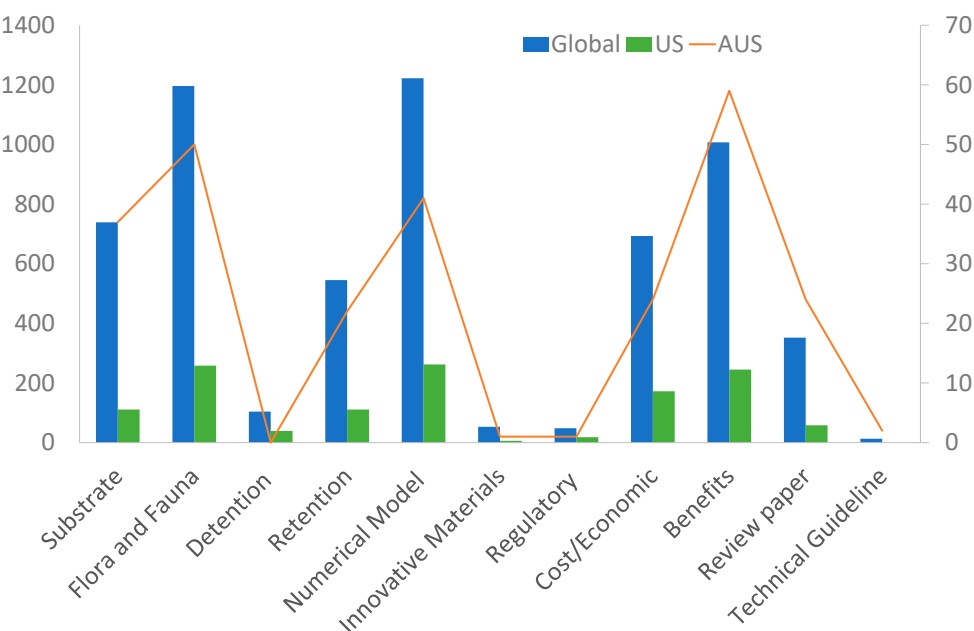

**Figure 7.** Research Trend Analysis.

By analysing the research trends, a few distinctive patterns can be noticed. Firstly, it can be seen that the US and global research trends nearly follow similar patterns with numerical models at the top of the list, followed by flora and fauna, benefits, substrate types, cost–benefit and retention. Limited research has been carried out on detention, innovative material, regulatory framework and technical guidelines. Australian research is more attentive firstly to benefits, secondly to flora and fauna, and thirdly to numerical models, with nil to very low existence on detention and innovative materials.

As evident from the analysis above, green roof development has received more attention and maturity on the effect side in contrast to the science, research and development side. This brings both good and bad news; the good news is that the problem green roofs are trying to solve is real and green roofs have the potential, whereas the bad news is that there is a significant knowledge gap on the foundation side, which is more pronounced in Australia in comparison to the global pattern.

As observed from the research gap analysis above, the majority of the research on green roofs have focused on their potential benefits, and there is lack of the regulatory framework and technical guidelines.

Williams et al. [71] proposed that green roof uptake in Australia has been impeded by a lack of scientific data available to evaluate their suitability to local conditions. Since this article was published, there has been very limited development in obtaining scientific data on green roofs. The recent review paper by Alim et al. [37] found the same limitations.

Further research on green roofs should be directed towards the development of software to analyse and design green roofs, the integration of green roofs with solar tiles, life cycle cost analysis, finding innovative materials and the multi-use of green roofs such as flood management, food production and recreational aspects.

## 9. Conclusions

With 56% of the world's population living in urban areas and the expectation of this percentage to grow, urbanisation is on the rise. Urbanisation has been identified as a key contributor to pluvial flash floods and the rapid degradation of urban and receiving water bodies. Although on-source management of urban runoff is considered the most effective way to manage such adverse effects, this gets constrained by the lack of available space in existing developed areas and new apartments. Green roofs are considered a key mitigation device, which can manage both stormwater quantity, quality and numerous

other benefits while satisfying the space constraints. Green roofs also contribute towards 12 out of 17 SDGs. Irrespective of these benefits, the uptake of green roofs as a potential stormwater management solution has been slow in many countries, such as Australia. A lack of regulatory framework, caused by the absence of experimental data and subsequent knowledge gaps establishing the effectiveness of green roofs and their subsequent evolutions, is a major cause behind this restraint.

Future research on green roofs and their subsequent evolutions should put a focus on gathering experimental data towards establishing a performance benchmark for the basics: detention, retention and water quality. Such data can be utilised towards developing standards and either the development of stand-alone software or integration into existing software. The contribution of green roofs towards the reduction of heat island effects needs further research, as urban temperature is increasing with time due to the rise in impervious areas and global warming. A multidisciplinary team is needed to effectively plan for greener cities in the future to make our cities more sustainable.

**Author Contributions:** M.A.R. conducted the study and drafted the manuscript, M.A.A. edited the manuscript and interpreted data, A.R. conceptualized, supervised the study and edited the manuscript, S.J. edited the manuscript and improved the writing. All authors have read and agreed to the published version of the manuscript.

**Funding:** The authors declare that no funding was received to carry out the research presented in this article.

**Institutional Review Board Statement:** Not applicable.

**Informed Consent Statement:** Not applicable.

**Data Availability Statement:** The datasets used in this study can be obtained from Scopus database.

**Acknowledgments:** The authors would like to acknowledge the support from Western Sydney University.

**Conflicts of Interest:** The authors declare that they have no conflict of interest.

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
