# Peer review of "Vegetated Roofs as a Means of Sustainable Urban Development: A Scoping Review"

_water, doi:10.3390/w14193188_

Round 1

Reviewer 1 Report

The manuscript is well organized and written.  I have no comment on the revision of the manuscript.

Author Response

Reviewer 1:

The manuscript is well organized and written.  I have no comment on the revision of the manuscript.

Authors’ response: Many thanks for your positive feedback. Nothing to address.

Reviewer 2 Report

The topic is very interesting, and it certainly has relevance. The research plan seems well done. The methodology is good, but it needs more robust information in the introduction, analysis, and discussion. However, the relevance to this particular journal should be enhanced in the entire paper with the considerations of scope and readership of the Journal. My major observations are mentioned below

1.  The introduction needs to be enhanced with more literature review. There are many urban development related studies published recently that need to bring the findings and methods used in those studies in the introduction section. Some of the documents are mentioned below found from the google scholar results are relevant to this study and published in different cities. Need to cite the below-mentioned paper appropriately and mention the research gap the study addressed. There are a few others which might be included for addressing the research gap.

Modelling the impacts of land use/land cover changing pattern on urban thermal characteristics in Kuwait

Predicting Microscale Land Use/Land Cover Changes Using Cellular Automata Algorithm on the Northwest Coast of Peninsular Malaysia

Assessment on controlling factors of urbanization possibility in a newly developing city of the Vietnamese Mekong delta using logistic regression analysis

Assessing the impacts of vegetation cover loss on surface temperature, urban heat island and carbon emission in Penang city, Malaysia

Classification of cities in Bangladesh based on remote sensing derived spatial characteristics

Application of the Optimal Parameter Geographic Detector Model in the Identification of Influencing Factors of Ecological Quality in Guangzhou, China

Retrieving spatial variation of aerosol level over urban mixed land surfaces using Landsat imageries: Degree of air pollution in Dhaka Metropolitan Area

Assessing and predicting land use/land cover, land surface temperature and urban thermal field variance index using Landsat imagery for Dhaka Metropolitan area

simulating the Relationship between Land Use/Cover Change and Urban Thermal Environment Using Machine Learning Algorithms in Wuhan City, China

Predicting the impacts of land use/land cover changes on seasonal urban thermal characteristics using machine learning algorithms

Monitoring the effects of vegetation cover losses on land surface temperature dynamics using geospatial approach in Rajshahi city, Bangladesh

Impact of vegetation cover loss on surface temperature and carbon emission in a fastest-growing city, Cumilla, Bangladesh

Modelling future land use land cover changes and their impacts on land surface temperatures in Rajshahi, Bangladesh.

Cellular Automata approach in dynamic modelling of land cover changes using RapidEye images in Dhaka, Bangladesh

Remote sensing approach to simulate the land use/land cover and seasonal land surface temperature change using machine learning algorithms in a fastest-growing megacity of …

Geospatial modelling of changes in land use/land cover dynamics using Multi-layer perceptron Markov chain model in Rajshahi City, Bangladesh

The operational role of remote sensing in assessing and predicting land use/land cover and seasonal land surface temperature using machine learning algorithms in Rajshahi …

2. What is the novelty of your work? Need to be enriched more

3. The interaction between results & discussion and conclusion need to be enhanced, which is missing. The information needs to be revised in the conclusion which is already discussed in the discussion section. Need to add new and important information only in the conclusion section which is not discussed previously.

4. Proofreading by a professional should be conducted to improve both language and organization quality. I recommend proofreading the paper from http://digonresearch.org/ and adding the proofreading certificate.

Author Response

Comments and Suggestions for Authors

The topic is very interesting, and it certainly has relevance. The research plan seems well done. The methodology is good, but it needs more robust information in the introduction, analysis, and discussion. However, the relevance to this particular journal should be enhanced in the entire paper with the considerations of scope and readership of the Journal. My major observations are mentioned below.

1.     The introduction needs to be enhanced with more literature review. There are many urban development related studies published recently that need to bring the findings and methods used in those studies in the introduction section. Some of the documents are mentioned below found from the google scholar results are relevant to this study and published in different cities. Need to cite the below-mentioned paper appropriately and mention the research gap the study addressed. There are a few others which might be included for addressing the research gap.

suggested papers are included in the revised manuscript.

Modelling the impacts of land use/land cover changing pattern on urban thermal characteristics in Kuwait

Predicting Microscale Land Use/Land Cover Changes Using Cellular Automata Algorithm on the Northwest Coast of Peninsular Malaysia

Assessment on controlling factors of urbanization possibility in a newly developing city of the Vietnamese Mekong delta using logistic regression analysis

Assessing the impacts of vegetation cover loss on surface temperature, urban heat island and carbon emission in Penang city, Malaysia

Classification of cities in Bangladesh based on remote sensing derived spatial characteristics

Application of the Optimal Parameter Geographic Detector Model in the Identification of Influencing Factors of Ecological Quality in Guangzhou, China

Retrieving spatial variation of aerosol level over urban mixed land surfaces using Landsat imageries: Degree of air pollution in Dhaka Metropolitan Area

Assessing and predicting land use/land cover, land surface temperature and urban thermal field variance index using Landsat imagery for Dhaka Metropolitan area

simulating the Relationship between Land Use/Cover Change and Urban Thermal Environment Using Machine Learning Algorithms in Wuhan City, China

Predicting the impacts of land use/land cover changes on seasonal urban thermal characteristics using machine learning algorithms

Monitoring the effects of vegetation cover losses on land surface temperature dynamics using geospatial approach in Rajshahi city, Bangladesh

Impact of vegetation cover loss on surface temperature and carbon emission in a fastest-growing city, Cumilla, Bangladesh

Modelling future land use land cover changes and their impacts on land surface temperatures in Rajshahi, Bangladesh.

Cellular Automata approach in dynamic modelling of land cover changes using RapidEye images in Dhaka, Bangladesh

Remote sensing approach to simulate the land use/land cover and seasonal land surface temperature change using machine learning algorithms in a fastest-growing megacity of …

Geospatial modelling of changes in land use/land cover dynamics using Multi-layer perceptron Markov chain model in Rajshahi City, Bangladesh

The operational role of remote sensing in assessing and predicting land use/land cover and seasonal land surface temperature using machine learning algorithms in Rajshahi …

Authors’ response: Thanks for your suggestions. The following relevant references are added in the revised manuscript (Introduction Section):

AlDousari, A. E., Kafy, A. A., Saha, M., Fattah, M. A., Almulhim, A. I., Al Rakib, A., ... & Rahman, M. M. (2022). Modelling the impacts of land use/land cover changing pattern on urban thermal characteristics in Kuwait. Sustainable Cities and Society, 86, 104107.

Rahaman, Z. A., Kafy, A. A., Faisal, A. A., Al Rakib, A., Jahir, D. M., Fattah, M., ... & Rahman, M. T. (2022). Predicting Microscale Land Use/Land Cover Changes Using Cellular Automata Algorithm on the Northwest Coast of Peninsular Malaysia. Earth Systems and Environment, 1-19.

Kafy, A. A., Al Rakib, A., Akter, K. S., Jahir, D. M. A., Sikdar, M. S., Ashrafi, T. J., ... & Rahman, M. M. (2021). Assessing and predicting land use/land cover, land surface temperature and urban thermal field variance index using Landsat imagery for Dhaka Metropolitan area. Environmental Challenges, 4, 100192.

Dey, N. N., Al Rakib, A., Kafy, A. A., & Raikwar, V. (2021). Geospatial modelling of changes in land use/land cover dynamics using Multi-layer perception Markov chain model in Rajshahi City, Bangladesh. Environmental Challenges, 4, 100148.

Kafy, A. A., Al Rakib, A., Fattah, M. A., Rahaman, Z. A., & Sattar, G. S. (2022). Impact of vegetation cover loss on surface temperature and carbon emission in a fastest-growing city, Cumilla, Bangladesh. Building and Environment, 208, 108573.

  1. What is the novelty of your work? Need to be enriched more

Authors’ response: Thanks for your suggestions. The following sentence is added in the Introduction section:

“The innovation in this study is to identify how the GI can contribute towards more sustainable urban stormwater management and how it can fulfil the sustainable development goals”. There is a section in the manuscript called “Innovation” (Section 5). 

  1. The interaction between results & discussion and conclusion need to be enhanced, which is missing. The information needs to be revised in the conclusion which is already discussed in the discussion section. Need to add new and important information only in the conclusion section which is not discussed previously.

Authors’ response: Thanks for the suggestion. The conclusion section is enhanced.

  1. Proofreading by a professional should be conducted to improve both language and organization quality. I recommend proofreading the paper from http://digonresearch.org/ and adding the proofreading certificate.

Authors’ response: The manuscript is fully edited by a native English speaker.

Reviewer 3 Report

The manuscript presents a review dealing with the vegetated roof as a mean of Sustainable Urban Development. In general, the topic falls within the aim and scope of Water Journal and of the special issue entitled: “Review Papers of Urban Water Management”. In addition to that, the English language is of high quality and the structure and the content of the work meets the requirements of a scientific publication. The authors however, should pay attention on the following issue and revise properly their work.  

1.       Figure 4 is of low quality. Please revise, and revise also the title of the figure to descried clearly the context of the 4 associated sub-figures.

Thus, the manuscript calls for revision.

Author Response

Comments and Suggestions for Authors

The manuscript presents a review dealing with the vegetated roof as a mean of Sustainable Urban Development. In general, the topic falls within the aim and scope of Water Journal and of the special issue entitled: “Review Papers of Urban Water Management”. In addition to that, the English language is of high quality and the structure and the content of the work meets the requirements of a scientific publication. The authors however, should pay attention on the following issue and revise properly their work.  

Authors’ response: Many thanks for your positive comments.

  1. Figure 4 is of low quality. Please revise, and revise also the title of the figure to descried clearly the context of the 4 associated sub-figures.

Authors’ response: Figure 4 was created using a bibliometric software named "CiteSpace". These diagrams could not be further optimised. We feel the diagram conveys the intended message very well.

Round 2

Reviewer 2 Report

The authors did all the Corrections mentioned by reviewers. 

Author Response

Thanks for your acceptance of the manuscript. There is no comment to address.